# CONTINUOUS CONVOLUTIONAL NEURAL NETWORK FOR NONUNIFORM TIME SERIES

## ABSTRACT

Convolutional neural network (CNN) for time series data implicitly assumes that the data are uniformly sampled, whereas many event-based and multi-modal data are nonuniform or have heterogeneous sampling rates. Directly applying regular CNN to nonuniform time series is ungrounded because it is unable to recognize and extract common patterns from the nonuniform input signals. Converting the nonuniform time series to uniform ones by interpolation preserves the pattern extraction capability of CNN, but the interpolation kernels are often preset and may be unsuitable for the data or tasks. In this paper, we propose the Continuous CNN (CCNN), which estimates the inherent continuous inputs by interpolation, and performs continuous convolution on the continuous input. The interpolation and convolution kernels are learned in an end-to-end manner, and are able to learn useful patterns despite the nonuniform sampling rate. Besides, CCNN is a strict generalization to CNN. Results of several experiments verify that CCNN achieves a better performance on nonuniform data, and learns meaningful continuous kernels.

## 1 INTRODUCTION

Convolutional neural network (CNN), together with recurrent neural network (RNN), is among the most popular deep learning architectures to process time series data. However, both CNN and RNN rest on the assumption that both the input and output data are sampled uniformly. However, many time-series data are event-based and thus not uniform in time, such as stock price (Gençay et al., 2001), social media data (Chang et al., 2016) and health care data (Johnson et al., 2016).

There are several easy solutions to adapt CNN to accommodate nonuniform time series. The first solution is to directly append the time stamps or time intervals to the input features, which are then fed into a regular CNN (Zhang et al., 2017). However, the problem is that, without the uniform sampling assumptions, the application of the regular CNN is ungrounded, and thus the performance is compromised. This is because one major justification of CNN is that the filters/kernels are able to extract useful patterns from input signals. But if the sampling rate varies, the traditional CNN will no longer be able to recognize the same pattern.

A second obvious solution is to transform the nonuniform time series to uniform by interpolation, and then feed the transformed signal to a regular CNN. This approach preserves CNN's ability to extract signal patterns despite the nonuniform sampling. However, simple interpolation schemes require preset interpolation kernels, which are not flexible and may not fit the signal or the task well. To sum up, most existing CNN-based remedies for nonuniform time series either cannot reasonably capture the signal patterns or are too inflexible to maximize the performance in a data-driven manner.

Motivated by these challenges, we propose Continuous CNN (CCNN), a generalization to CNN for nonuniform time series. CCNN estimates the implicit continuous signal by interpolation, yet performs continuous convolution on the continuous signal. As a result, CCNN is capable of capturing the useful patterns in the implicit input signal, which is of nonuniform sampling rate or naturally has an uneven time interval. Furthermore, the interpolation and convolution kernel functions are not preset, but rather learned in an end-to-end manner, so that the interpolation is tailored for each task. Finally, we show that CCNN and CNN are equivalent in terms of representation power under a uniform sampling rate. As shown in section 5, CCNN can achieve much better performance than the state-of-the-art systems on non-uniform time series data.

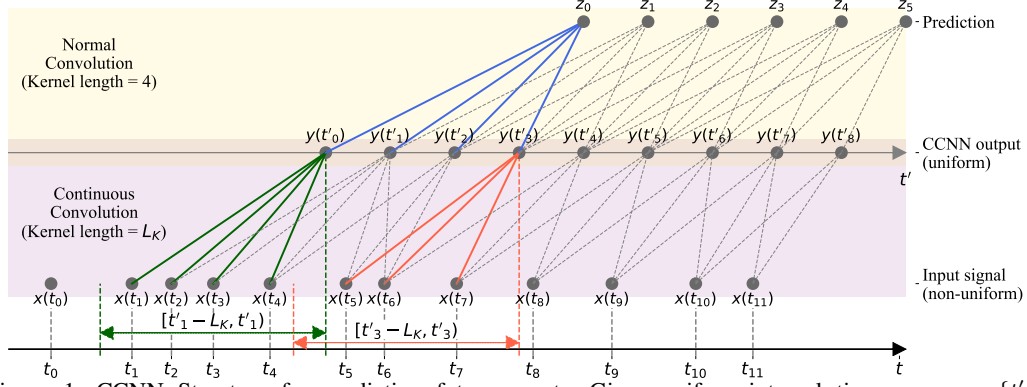

Figure 1: CCNN Structure for predicting future events. Given uniform-interval time sequence $\{t'_i\}$, CCNN layer performs both interpolating non-uniformly sampled signal sequence $\{x(t_i)\}$ to $\{\hat{x}(t'_i)\}$ and convolution($\{y(t'_i)\}$). Now that $\{y(t'_i)\}$ is uniformly resampled, normal convolution layer can be applied.

The proposed CCNN can also be well-combined with the temporal point process (TPP). TPP is a random process of event occurrence, which is capable of modeling nonuniform time intervals. However, most existing TPPs require inflexible preset parameterization. CCNN is able to expand the power of TPP by replacing the modeling of the history contribution with a CCNN module.

## 2 RELATED WORKS

There are some research efforts of adapting RNN for nonuniform series. Some works (Pearlmutter, 2008; Funahashi & Nakamura, 1993; Cauwenberghs, 1996) use continuous-time dynamical system methods to design RNN structures. Phased-LSTM (Neil et al., 2016) and Time-LSTM (Zhu et al., 2017) introduce a time gate that passes the data at a certain frequency. Similar ideas can be found in Clockwork RNN (Koutnik et al., 2014) and DilatedRNN (Chang et al., 2017). (Mei & Eisner, 2017) and (Du et al., 2016) explicitly model the sequence as a temporal point process and utilize RNN structure to encode the sequence history.

For non-neural network-based approaches, the probabilistic generation process of both events and its timestamps is assumed. (Liu & Hauskrecht, 2016) deals with irregularly sampled time series by direct value interpolation and estimates the model via EM algorithm. (Wang et al., 2016; Du et al., 2015) base their model on the Hawkes process and estimate via conditional gradient algorithm.

The proposed CCNN is well supported by works on spiking neural networks (SNN) (Maass, 1997), which mimic how human brains process information. The inputs to SNNs are spike chains with nonuniform intervals that carry information. An important class SNNs (Eliasmith & Anderson, 2004; Tapson & van Schaik, 2013; Tapson et al., 2013) convolves the input with continuous kernel functions, which is similar to the key step of CCNN. However, CCNN differs from SNN in two aspects. First, for SNN, the input information resides in time intervals, not in the inputs values; the goal of the SNN convolution is to extract time interval information. In contrast, the input information for CCNN resides in input values, not time intervals; the goal of the continuous convolution is to remove the interference of nonuniform sampling. Second, CCNN learns the kernel functions in a data-driven manner, whereas SNN employs predefined kernel functions.

Nonuniform time series processing is related to the task of point set classification, where the input is a set of points nonuniformly distributed in the $\mathbb{R}^d$ space ($d = 3$ in most cases). Several existing methods directly work on the coordinates of the points (Qi et al., 2017a;b). Some alternative approaches turn the point sets into graphs by establishing edge connections among nearest neighbors (Shen et al., 2017; Wang et al., 2018). The graph approaches utilize distance information to some degree, but the distance information is quantized. CCNN, on the other hand, make full use of the time interval information.

## 3 THE CCNN ALGORITHM

Our problem is formulated as follows. Given a nonuniform input sequence $x(t_1), x(t_2), \cdots, x(t_N) \in \mathcal{X}_{\text{in}}$, where the input time stamps $t_n \in \mathcal{T}_{\text{in}}$ can be distributed nonuniformly, our goal is to design a continuous convolutional layer that can produce output for any *arbitrary* output time $t$, $y(t)$.

The proposed CCNN solves the problem via two steps:

(1) interpolation to recover the continuous signal $\hat{x}(t)$;

(2) *continuous* convolution on $\hat{x}(t)$.

Furthermore, rather than applying a preset interpolation, CCNN learns the interpolation kernel and the convolution kernel in an end-to-end manner. The following two subsections elaborate on the two steps respectively. The channel dimension and input dimension are set to one for simplification.

## 3.1 INTERPOLATION

CCNN reconstructs the underlying continuous input signal, $\hat{x}(t)$, by interpolating among nonuniform input samples.

$$\hat{x}(t) = \sum_{i=1}^{N} x(t_i) I(t - t_i; \mathcal{T}_{\text{in}}, \mathcal{X}_{\text{in}}) + \varepsilon(t; \mathcal{T}_{\text{in}}, \mathcal{X}_{\text{in}}) \tag{1}$$

where the first term is the interpolation term, and $I(\cdot)$ is the *interpolation kernel*; the second term is the *error correction* term. For the first term, a form analogous to the Parzen window approach Parzen (1962) is used. Many interpolation algorithms can be expressed in this form (refer to Appendix A), illustrated in Fig. 6. Considering the versatility of $I(\cdot)$, the interpolation algorithms representable by Eq. (1) are vast. The error correction term, $\varepsilon(\cdot)$, are assumed to be determined by the input output time stamps and input values, hence its arguments include $t$, $\mathcal{T}_{\text{in}}$ and $\mathcal{X}_{\text{in}}$.

## 3.2 CONTINUOUS CONVOLUTION

Analogous to a standard CNN layer, after the continuous input is estimated by interpolation, the CCNN layer performs a continuous convolution to produce the final output.

$$y(t) = \hat{x}(t) * C(t) + b \tag{2}$$

where $*$ denotes continuous convolution, $C(t)$ denotes the *convolution kernel*, and $b$ denotes bias.

Unfortunately, $I(\cdot)$, $\varepsilon(\cdot)$ and $C(\cdot)$ are not individually identifiable. To see this, we combine Eqs. (1) and (2).

$$y(t) = \sum_{i=1}^{N} x(t_i) \underbrace{[I(t - t_i; \mathcal{T}_{\text{in}}, \mathcal{X}_{\text{in}}) * C(t)]}_{\text{collapsed kernel function}} + \underbrace{[\varepsilon(t; \mathcal{T}_{\text{in}}, \mathcal{X}_{\text{in}}) * C(t) + b]}_{\text{collapsed bias function}}$$

$$= \sum_{i=1}^{N} x(t_i) K(t - t_i; \mathcal{T}_{\text{in}}, \mathcal{X}_{\text{in}}) + \beta(t; \mathcal{T}_{\text{in}}, \mathcal{X}_{\text{in}}) \tag{3}$$

where $K(t; \mathcal{T}_{\text{in}}, \mathcal{X}_{\text{in}})$ is the collapsed kernel function, representing the combined effect of interpolation and convolution; $\beta(t; \mathcal{T}_{\text{in}}, \mathcal{X}_{\text{in}})$ is the collapsed bias function, representing the combined effect of error correction and convolution.

Eq. (3) shows that learning the interpolation and convolution kernels and errors is now simplified into learning the collapsed kernel and bias functions. Once these two functions are learned, the final output can be readily computed using Eq. (3). The next section will explain how CCNN is structured to learn these functions in an end-to-end manner.

## 4 THE CCNN STRUCTURE

Following the discussion in Sec. 3, a CCNN layer is divided into three parts: the kernel network learning the collapsed kernel function, the bias network learning the collapsed bias function, and the main network producing the final output using Eq. (3).

## 4.1 THE KERNEL NETWORK

The basic idea of the kernel network is to represent the kernel function using a neural network, based on the fact that a neural network can represent any function given enough layers and nodes (Hornik, 1991). In order to regularize the complexity, a few assumptions on $K(\cdot)$ are introduced:

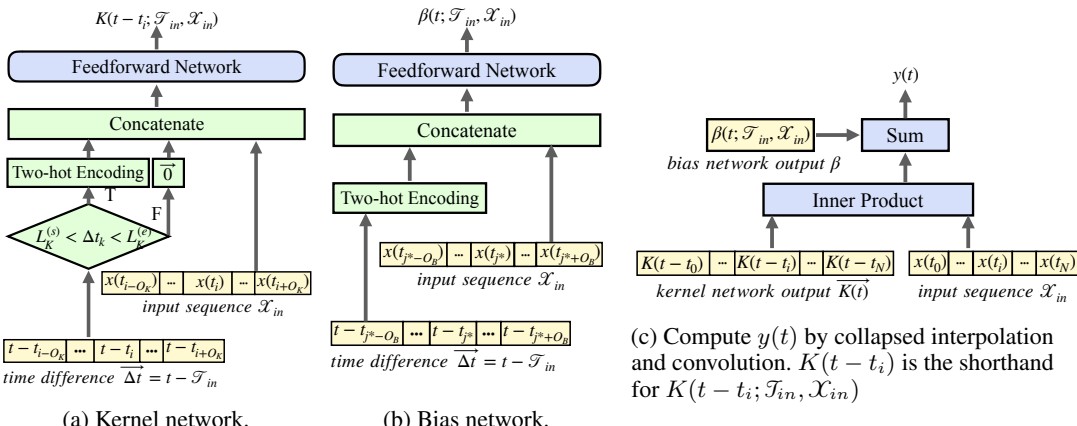

(a) Kernel network.  (b) Bias network.

(c) Compute $y(t)$ by collapsed interpolation and convolution. $K(t - t_i)$ is the shorthand for $K(t - t_i; \mathcal{T}_{in}, \mathcal{X}_{in})$

Figure 2: CCNN structure.

**Stationarity and Finite Dependency:** The dependency of $K(\cdot)$ on $\mathcal{T}_{in}$ is relative to the output time $t$, and is constrained to among the adjacent time stamps, i.e.

$$K(t - t_i; \mathcal{T}_{in}, \mathcal{X}_{in}) = K(\{t - t_{i \pm k}, x(t_{i \pm k})\}_{k=0:O_K}) \tag{4}$$

where $\{t - t_{i \pm k}, x(t_{i \pm k})\}_{k=0:O_K}$ denotes the set of $t - t_{i \pm k}$ and $x(t_{i \pm k})$ where $k$ runs from 0 to $O_K$, and $O_K$ is the order of the kernel network. Notice that the examples in Eqs. (14)-(16) and many other interpolation kernels still satisfy this assumption.

**Finite Kernel Length:** The collapsed kernel function has finite length.

$$K(t - t_i; \mathcal{T}_{in}) = 0, \forall |t - t_i| > L_K \tag{5}$$

where $L_K$ is the kernel length. This assumption implies the interpolation and the convolutional kernels both have finite length. While many interpolation kernels do have finite length (e.g. Eqs. (14) and (15)), others do not (e.g. Eq. (16)). Nevertheless, most infinite-length interpolation kernels, including Eq. (16), have tapering tails, and thus truncation on both sides still provides good approximations. Regarding the convolutional kernel, the finite length assumption naturally extends from the standard CNN.

Fig. 2(a) shows the kernel network structure, which is a feedforward network. According to Eq. (4), the inputs are $(\{t - t_{i \pm k}, x(t_{i \pm k})\}_{k=0:O_K})$. The output represents the kernel function, which is forced to be zero when $|t - t_i| > L_K$. To reduce learning difficulties, the time differences are fed into an optional two-hot encoding layer, which will be discussed later in details.

## 4.2 THE BIAS NETWORK

For the bias network, a similar stationarity and finite dependency assumption is applied as follows.

$$\beta(t; \mathcal{T}_{in}, \mathcal{X}_{in}) = \beta(\{t - t_{j^* \pm k}, x(t_{j^* \pm k})\}_{k=0:O_B}), \tag{6}$$

where $t_{j^*}$ is the closest input time stamp to output time $t$:

$$t_{j^*} = \underset{t_j \in \mathcal{T}_{in}}{\operatorname{argmin}} |t_j - t| \tag{7}$$

and $O_B$ denotes the order of the bias network. The only difference from Eq. (4) is that the closest input time stamp, $t_{j^*}$, is chosen as a reference on which the time difference and the adjacent input time stamps are defined, because the major argument of the bias function is the output time itself, $t$, instead of the input-output time difference $t - t_i$. Fig. 2(b) shows the bias network, which is also a feedforward network.

## 4.3 CAUSAL SETTING

For causal tasks, current output should not depend on future input, and therefore the $t - t_{i \pm k}$ terms that are greater than 0, as well as the corresponding $x(t_{i \pm k})$, are removed from Eq. (4). Similarly, $t - t_{j^* \pm k}$ that are greater than 0, as well as the corresponding $x(t_{j^* \pm k})$, are removed from Eq. (6). Also, the condition bound in Eq. (5) is replaced with $t - t_i > L_K$ or $t - t_i < 0$.

Figure 3: Two-hot encoding. Each cross on the 1-D axis denotes a value of $\Delta t$. The stem plot above shows its two-hot vector. Assuming $d = 5$, the left plot shows when $\Delta t = \pi_{k-1} + 0.4\delta$, the two-hot encoding of $\Delta t$ is [0, 0.6, 0.4, 0, 0]. The right plot shows when $\Delta t = \pi_{k-1}$, the encoding is [0, 1, 0, 0, 0]

### 4.4 TWO-HOT ENCODING

The kernel and bias functions can be complicated functions of the input times, so model complexity and convergence can be serious concerns. Therefore, we introduce a two-hot encoding scheme for the input times, which is an extension to the one-hot scheme, but which does not lose information.

Denote the time difference value to be encoded as $\Delta t$. Similar to one-hot, the two-hot encoding scheme partitions the range of $\Delta t$ into $D - 1$ intervals, whose edges are denoted as $\pi_1, \pi_2, \cdots, \pi_d$. However, rather than having a length-$D - 1$ vector representing the intervals, two-hot introduces a length-$D$ vector representing the edges. When $\Delta t$ falls in an interval, the two elements corresponding to its two edges are lit. Formally, denote the encoded vector as $g$, and suppose $\Delta t$ falls in interval $[\pi_k, \pi_{k+1})$. Then

$$g_k = \frac{\pi_{k+1} - \Delta t}{\delta}, g_{k+1} = \frac{\Delta t - \pi_k}{\delta}; g_l = 0, \forall l \notin \{k, k+1\} \tag{8}$$

where $\delta = \pi_k - \pi_{k-1}$ denotes the interval width (all the intervals are set to have equal width); $g_k$ denotes the $k$-th element of $g$. Fig. 3 gives an intuitive visualization of the encoding process.

As an example explanation of why two-hot helps, it can be easily shown that a one-layer feed-forward network can only learn a linear function (a straight line) without any encoding, but a piecewise constant function with one-hot encoding, and yet a piecewise linear function with two-hot encoding.

### 4.5 COMBINING WITH TEMPORAL POINT PROCESSES

For tasks like predicting the time interval till the next event, the output of CCNN will be the predicted probability distribution of the time interval, which requires a good probabilistic model characterizing the likelihood of these intervals. Temporal point process (TPP) is a popular and general model for the timestamps of the random processes $\{x(t_i), t_i\}$ whose time intervals are nonuniform. It turns out that CCNN can be well combined with TPPs in modeling the time interval prediction task, in a similar way to Du et al. (2016).

A TPP is parameterized by $\lambda^*(t)$, which depicts the rate of the probability of the event occurrence. Formally

$$\lambda^*(t)dt = Pr\left(\text{Event } i \text{ happens in } [t, t+dt] \left| \bigcup_{j<i}\{x(t_j), t_j\}\right.\right) \tag{9}$$

It can be shown that the probability density function (PDF) of an event happening at time $t$ conditional on the history of the events $\bigcup_{j\leq i-1}\{x(t_j), t_j\}$ can be expressed as

$$f^*(t) = \lambda^*(t)\exp\left(-\int_{t_{i-1}}^t \lambda^*(\tau)d\tau\right). \tag{10}$$

Rather than applying some preset functional form for $\lambda^*(t)$ as in conventional TPPs, we propose to use a CCNN to model $\lambda^*(t)$ as follows. First, we pass the historical time series to a CCNN to learn a history embedding

$$h_{i-1} = \text{CCNN}\left(\bigcup_{j\leq i-1}\{x(t_j), t_j\}\right), \tag{11}$$

where CCNN $(\cdot)$ is just a functional abstraction of CCNN. Then $\lambda^*(t)$ is obtained by combining the history information and the current time information as follows

$$\lambda^*(t) = \exp(vh_{i-1} + w(t - t_{i-1}) + b), \tag{12}$$

| Alg. | Sine | MG | Lorenz |
|---|---|---|---|
| CNN | 46.0 (8.22) | 12.8 (3.92) | 9.90 (3.33) |
| CNNT | 20.2 (7.65) | 3.50 (1.29) | 5.97 (2.41) |
| CNNT-th | 8.44 (4.58) | 3.00 (1.21) | 8.37 (3.24) |
| ICNN-L | 1.13 (0.87) | 0.97 (0.53) | 5.81 (2.78) |
| ICNN-Q | 0.75 (0.65) | 0.83 (0.46) | 5.08 (2.59) |
| ICNN-C | 0.72 (0.83) | 0.72 (0.42) | 4.22 (2.27) |
| ICNN-P | 20.5(6.43) | 1.95(0.79) | 8.50(3.32) |
| ICNN-S | 17.2(5.57) | 3.51(1.36) | 8.20(3.31) |
| RNNT | 36.1(12.9) | 8.15(3.32) | 13.4(3.95) |
| RNNT-th | 19.5(6.48) | 8.48(3.11) | 13.9(4.36) |
| CCNN | 0.88 (0.61) | 2.46 (0.89) | 3.93 (1.73) |
| CCNN-th | **0.42** (0.36) | **0.53** (0.97) | **3.25** (1.67) |

Table 1: Mean squared error of prediction on simulated data ($\times 10^{-2}$).

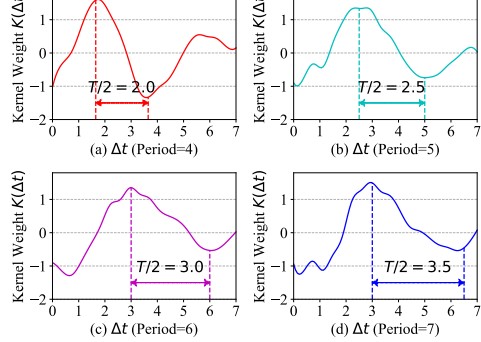

Figure 4: The learned continuous kernel function on the sine, as functions of $\Delta t = t - t_i$.

where $\boldsymbol{v}$, $w$ and $b$ are trainable parameters. Combining Eqs. (10) and (12), we can obtain a closed-form expression for $f^*(t)$

$$f^*(t) = \exp\left(\boldsymbol{vh}_{i-1} + w(t - t_{i-1}) + b + \frac{1}{w}\Big(\exp(\boldsymbol{vh}_{j-1} + b) - \exp(\boldsymbol{vh}_{j-1} + w(t - t_{j-1}) + b)\Big)\right). \tag{13}$$

By maximizing this likelihood on the training data, we can estimate the conditional distribution of the time intervals. To obtain a point estimate of the time interval till the next event, we compute the expectation under Eq. (13) numerically.

Du et al. (2016) also applies the same approach, but the history embedding $\boldsymbol{h}_{i-1}$ is computing by a regular RNN. CCNN, with its improved processing of nonuniform data, is expected to produce a better history embedding, and thereby a better estimate of the time intervals.

### 4.6 Summary and Generalization

Fig. 2 illustrates the structure of a CCNN layer. To sum up, the kernel network and bias network learn the continuous kernel and bias as functions of $t$, $\mathcal{T}_{\text{in}}$ and $\mathcal{X}_{\text{in}}$. The main network applies these functions to produce the output according to Eq. (3). The hyperparameters include $O_K$ (Eq. (4)), $L_K$ (Eq. (5)), $O_B$ (Eq.(6)) and $\delta$ (Eq. (8)).

It is worth highlighting that CCNN not only accommodates arbitrary input timestamps, it can also produce output at any output timestamps, by simply adjusting the value of $t$ in Eqs. (3), (4) and (6). So a CCNN layer can accept input at a set of timestamps, and produces output at a different set of timestamps, which is very useful for resampling, interpolation, and continuous sequence prediction.

When the inputs $x(t_i)$ and output $y(t)$ need to be multidimensional, according to Eq. (3), $K(\cdot)$ and $\beta(\cdot)$ become vectors or matrices with matching dimensions. Therefore, we simply need to adapt the output of the kernel and the bias networks from scalars to vectors or vectorized matrices. Also, a multi-layer CCNN can be constructed by stacking individual CCNN layers, with the input timestamps of a layer matching the output timestamps of its previous layer.

## 5 Evaluation

In this section, CCNN is evaluated on a set of prediction tasks on both simulated and real-world data.

### 5.1 Predicting Signal Value on Simulated Data

The prediction task predicts the next sample $x(t_{N+1})$, given the previous nonuniform samples $x(t_1) \cdots x(t_N)$.

**Datasets** The synthetic datasets are generated by unevenly sampling from three standard time series: Sine, Mackey-Glass(MG) and Lorenz. The details of the time series are introduced in Appendix C.1

**Baselines** The following algorithms are compared:

• *CCNN:* The proposed algorithm. The first layer is a CCNN layer which takes both the sampling time intervals and the signal sequence. After the CCNN layer, the sequence is resampled onto a

uniform time interval. For predicting future label, signal value, and interval, the CCNN is configured using only past value, i.e. CCNN kernel has non-zero value only when $t' - L_K < t_i < t'$, shown in Fig. 1. The time information is either two-hot encoded (Adams et al., 2010) (CCNN-th) or not encoded (CCNN).

• *CNN:* data are directly fed into a regular CNN, with no special handling of nonuniform sampling.

• *CNNT:* The sampling time intervals are appended to the input data, which are fed to a regular CNN. The time information is either two-hot encoded (CNNT-th), or not encoded (CNNT).

• *ICNN:* data are interpolated to be uniform before being fed to a regular CNN. Piecewise Constant (ICNN-P), linear (ICNN-L), quadratic (ICNN-Q), cubic spline (ICNN-C) and sinc (ICNN-S) interpolation algorithms are implemented.

• *RNNT:* the sampling time intervals are appended to the input data, then are fed into a vanilla RNN. The time information is either two-hot encoded (RNNT-th), or not encoded (RNNT).

All the networks have two layers with ReLU activations in the hidden layers and no activations in the output layer. For CNN, ICNN, and CNNT, the convolution kernel length of each layer is set to 7. For ICNN, the input signal is interpolated at timestamps $t_{N+1} - k, k = 1, \cdots, 13$ to form a uniform sequence before feeding into two-layers regular CNN. For CCNN, the output time stamps of the first layer are $t_{N+1} - k, k = 1, \cdots, 13$. The kernel length $L_K = 3$. Since its input is uniform, the second layer of CCNN is a regular convolutional layer, with kernel length 7. These configurations ensure that all the neural networks have the same *expected* receptive field size of 13.

The rest of the hyperparameters are set such that all the networks have comparable number of parameters, as in Appendix C.2. All the networks are trained with Adam optimizer Kingma & Ba (2014) and mean squared error (MSE) loss. The training batch size is 20. The number of training steps is determined by validation. The validation set size is 10,000.

**Results and Analysis** Table 1[1] lists the MSEs. There are three observations. **First**, CCNN-th outperforms the other baselines in terms of prediction accuracy. Notice that the number of convolution channels of CCNN are significantly smaller than most of the other baselines, in order to match the number of parameters. Nevertheless, the advantage in properly dealing with the nonuniform sampling still offsets the reduction in channels in most tasks. **Second**, interpolation methods (ICNNs and CCNN) generally outperform the other baselines, particularly CNNT. This again shows that interpolation is more reasonable for dealing with nonuniform time series than simply appending the time intervals. Furthermore, preset interpolation algorithms (ICNNs) can rarely match CCNN that has the flexibility to learn its own interpolation kernel. **Third**, two-hot encoding usually improves performance. Again, there are fewer channels with two-hot encoding in order to match model complexity, but the advantage of two-hot encoding still stands out.

**Kernel Analysis** In order to visualize the learned continuous kernels $K(t - t_i; \mathcal{T}_{in}, \mathcal{X}_{in})$, we set the CCNN network has the same configuration except CCNN filter number is 1, and is trained separately with nonuniformly sampled ($\lambda = 1$) sine signals with $T = 4, 5, 6, 7$. The learned continuous kernel function is quite interpretable. Each kernel is a sine-like function with estimated period equaling the underlying signal period shown in Fig. 4

## 5.2 PREDICTING TIME INTERVALS TO NEXT EVENT

We then evaluate CCNN on real-world data to to predict the time intervals to the next event.

**Datasets and Configurations** Four time series datasets with nonuniform time intervals are introduced, *i.e.* NYSE, Stackoverflow[2], MIMIC ((Johnson et al., 2016)) and Retweet ((Zhao et al., 2015)). Detailed information are provided in Appendix C.3. The input sequence is the timestamps and the one-hot encoded types of a series of events. The task is to predict the time interval until the next event of a specific type, given the previous events. As mentioned in Sec. 4.5, following the design in (Du et al., 2016), the input sequence is assumed to be generated via an underlying marked TPP, where the time stamps follow a TPP, and the marker, *i.e.* the type of the event, is generated from a multinomial distribution conditioned on history. The output of the networks is a condition intensity function $\lambda^*(t)$

---

[1]Hereafter, in tables, the numbers in parentheses show standard deviation.
[2]https://archive.org/details/stackexchange

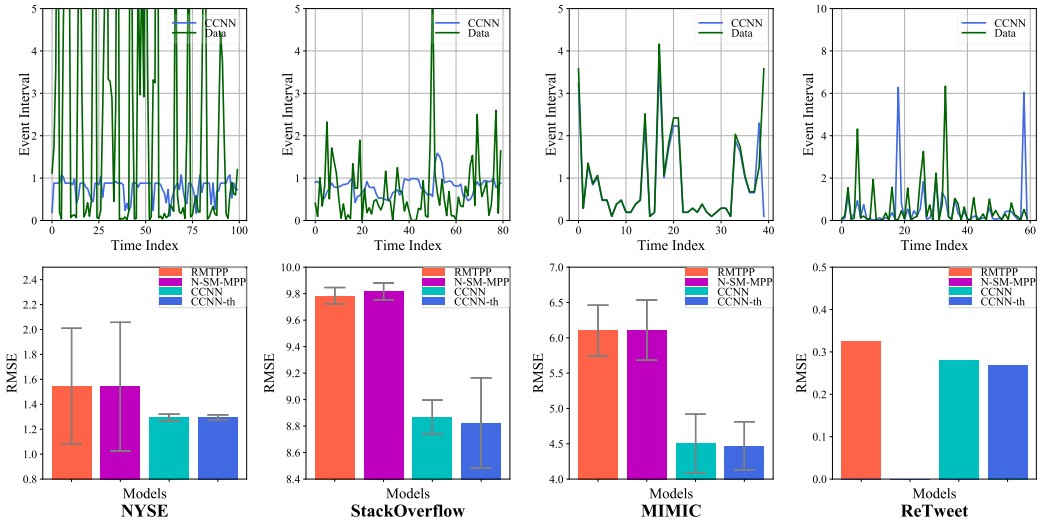

Figure 5: Example predicted time intervals, which is the expectation over Eq. (10) (upper) and RMSE (lower) on the predicting time interval to next event (section 5.2). Standard deviation is calculated among 5 train-test-validation splits. The ReTweet dataset has only one split so no standard deviation is reported. N-SM-MPP did not report RMSE on retweet dataset.

as in Eq. (12). The loss function is the log-likelihood of the training set as in (Du et al., 2016). As the model prediction, the expected duration is computed numerically from the estimated conditional distribution. The evaluation metric is the MSE of the expected duration and the actual duration to the next event. Configurations of all the models are the same as in previous experiments except that: the one-hot encoded event types, $\{x(t_i)\}$, are first passed through an embedding layer, which is a $1 \times 1$ convolutional layer with a channel dimension of 8, and the resulting embedded vectors are then fed into the networks.

This task is a causal task, where current output should not depend on future input. Therefore, the CCNN configuration is adapted to the causal setting, as discussed in section 4.3.

**Baselines** We benchmark with two baselines specialized for this type of tasks, Recurrent Marked Temporal Point Process (RMTPP) (Du et al., 2016) and N-SM-MPP (Mei & Eisner, 2017). N-SM-MPP is the current state-of-the-art deep learning method. For NYSE, StackOverflow and MIMIC, we directly compare to the results (Mei & Eisner, 2017) reported, and re-implemented RMTPP to benchmark ReTweet. Configurations for CCNNs are provided in Appendix C.4.

**Results and Analysis** Fig. 5 shows the estimated event interval (expectation of Eq. (10)) and RMSEs. The upper plots show that the predicted interval aligns well with ground truth and result in smaller RMSE in the MIMIC and Retweet dataset. In NYSE and StackOverflow, though the groud truth shows extremely fluctuation on event intervals and CCNNs fail to predict accurately as in MIMIC and ReTweet, the predicted interval still tends to capture the increase and decrease trend. The lower plots compare the RMSE with the baselines. CCNN algorithms outperformed two baselines in all datasets. There is a slight advantage of CCNN-th over CCNN, which verifies the effectiveness of two-hot encoding.

## 5.3 Additional Experiments

Two additional experiments on real-world data, which are predictions on Data Market and interpolation on speech, are presented in Appendix D.

## 6 Conclusion

In this paper, we have introduced CCNN for nonuniform time series with two takeaways. First, interpolation before continuous convolution is shown to be a reasonable way for nonuniform time series. Second, learning task-specific kernels in a data-driven way significantly improves the performance. There are two promising directions. First, we have focused on 1D convolution, but this framework can be generalized to multi-dimensional nonuniform data. Second, while the computational complexity is similar for CCNN and CNN, the runtime of the former is much longer, because of the lack of parallelization. Fast implementation of CCNN is thus another research direction.

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

## A   INTERPOLATION KERNELS EXAMPLES

• *Piecewise Constant Interpolation:*

$$I(t - t_i; \mathcal{T}_{\text{in}}, \mathcal{X}_{\text{in}}) = \mathbb{1}[0 < t - t_i \leq t_{i+1} - t_i] \qquad (14)$$

where $\mathbb{1}[\cdot]$ denotes the indicator function.

• *Linear Interpolation:*

$$I(t - t_i; \mathcal{T}_{\text{in}}, \mathcal{X}_{\text{in}}) = \begin{cases} \frac{t - t_{i-1}}{t_i - t_{i-1}} & \text{if } 0 < t - t_{i-1} < t_i - t_{i-1} \\ \frac{t_{i+1} - t}{t_{i+1} - t_i} & \text{if } 0 < t_{i+1} - t < t_{i+1} - t_i \\ 0 & \text{otherwise.} \end{cases} \qquad (15)$$

• *Sinc Interpolation:*

$$I(t - t_i; \mathcal{T}_{\text{in}}, \mathcal{X}_{\text{in}}) = a \sin\left(\frac{\pi(t - t_i)}{a}\right) / (t - t_i). \qquad (16)$$

These examples are illustrated in Fig 6. Notice that in Eqs. (14) and (15), the interpolation kernels depend not only on $t - t_i$, but also on adjacent input times, $t_{i+1}$ and/or $t_{i-1}$, and hence all the input times $\mathcal{T}_{\text{in}}$ are put in the argument. In some nonlinear interpolations, the interpolation kernel is also affected by the input values $\mathcal{X}_{\text{in}}$.

## B   REPRESENTATION POWER ANALYSIS

In this section, we study how the proposed CCNN layer relates to and compares against two CNN baselines in terms of representation power. The first baseline is simply a regular convolutional layer, and the second baseline is a convolutional layer with input time intervals, $\Delta t_i$, appended to the input features, which we will call CNNT throughout.

### B.1   CASE 1: OUTPUT TIMESTAMPS SAME AS INPUT

This subsection intuitively explains why CCNN has a superior representation power to CNNT. Suppose the output timestamps are the same as input timestamps, i.e. $t \in \mathcal{T}_{\text{in}}$. Then, combining Eq. (3) with Eqs. (4)-(6), we have

$$y(t_j) = \sum_i x(t_i) K(t_j - t_i; \mathcal{T}_{\text{in}}, \mathcal{X}_{\text{in}})$$
$$+ \beta(0; \{t_j - t_{j \pm k}, x(t_{j \pm k})\}_{k=0:O_B}). \qquad (17)$$

In contrast, for a CNNT layer, if we separate the convolution on time interval from the rest

$$y(t_j) = \sum_i x(t_i) K_{j-i} + \left[\sum_i (t_j - t_{j-1}) K'_{j-i} + b\right]. \qquad (18)$$

The second term of Eq. (17) represents a feed-forward network on $\{t_j - t_{j \pm k}, x(t_{j \pm k})\}_{k=0:O_B}$, whereas the second term of Eq. (18) can be regarded as a one-layer feed-forward network on

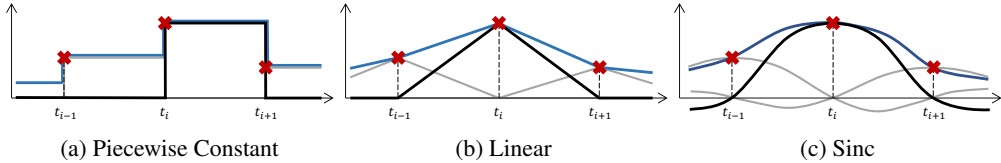

| (a) Piecewise Constant | (b) Linear | (c) Sinc |

Figure 6: Illustration of interpolation kernels. The red crosses denote the input data samples. The black line shows the interpolation kernel for $x(t_i)$; the gray lines show the kernels for the other two points. The blue line shows the interpolated result.

$\{t_j - t_{j-1}\}$, which is equivalent to a one-layer feedforward network on $\{t_j - t_{j\pm k}\}_{k=1:O_B}$. In other words, appending the time interval feature to the convolution layer is only a weak version of the CCNN bias network.

Yet a more fundamental disadvantage of CNNT lies in the first term, where the convolution kernel of CNNT, $K_{j-i}$, does not depend on the actual time difference, but the order in which the sample arrives. This makes CNNT very vulnerable to the sampling variations. CCNN, on the other hand, has a more robust kernel function for nonuniform data.

### B.2    CASE 2: UNIFORM TIMESTAMPS

Both CNNT and CCNN have the same representation power as CNN under uniform sampling rate, and thus both are strict generalizations to CNN. For CNNT this is trivial because the second term of Eq. (18) reduces to a constant. For CCNN, we have the following theorem.

**Theorem 1.** *Let $\mathcal{F}$ be the set of all functions that can be represented by a CCNN layer, and $\mathcal{G}$ be the set of all functions that can be represented by a $1 \times W$ convolutional layer. Then $\mathcal{F} = \mathcal{G}$, if the following conditions hold:*

*1. the input and output timestamps are uniform and the same, i.e. $\Delta t_i = \Delta t, \forall i$.*

*2. The two-hot encoding interval boundaries are at multiples of $\Delta t$, i.e. $\pi_k = k \Delta t$.*

*3. The dimension of the two-hot vector of CCNN is no smaller than the CNN kernel length, i.e. $D \geq W$.*

*4. If the kernel network has hidden layers, the hidden node dimension is no smaller than $W$.*

*5. CCNN and CNN have the same receptive field size, i.e. $2L_K = W\Delta t$.*

*6. The kernel and bias networks do not depend on $\mathcal{X}_{in}$.*

*Proof to Theorem 1.* We will only consider the 1D case. The generalization to multi-dimensional cases is straightforward. The regular $1 \times W$ CNN performs the following operation to generate the output sequence

$$y(\tau_k) = \sum_{i=k-(W-1)/2}^{k+(W-1)/2} x(t_i)K_{k-i} + b \tag{19}$$

where $\left\{K_{-(W-1)/2}, \cdots, K_{(W-1)/2}\right\}$ and $b$ are trainable parameters. Here we implicitly assume that $W$ is odd. We will prove the theorem by primarily utilizing the correspondence between Equations (17) and (19).

• $\mathcal{F} \subset \mathcal{G}$:

$\forall K(\{t_j - t_{j+k}\}_{k=\pm 0:\pm O_K})$ represented by the kernel network, and $\forall \beta(\{t_j - t_{j+k}\}_{k=1:O_B})$ represented by the bias network, and thereby $\forall f \in \mathcal{F}$.

By Eq. (4) and Cond. 1, the arguments of the kernel function $K(\cdot)$, namely the input to the kernel network, can only be a set of consecutive multiples of $\Delta t$, i.e.

$$K(\{(w+k)\Delta t\}_{k=\pm 0:\pm O_K}) \tag{20}$$

where $w$ is an integer.

Moreover, from Eq. (5) and Cond. 5, $K(\cdot)$ is non-zero iff $w$ lies in the interval $[-(W-1)/2, (W-1)/2]$.

Similarly, from Eq. (17) and Cond. 1, the arguments of the bias function $\beta(\cdot)$ can only take one set of values:

$$\beta(\{k\Delta t\}_{k=\pm 1:\pm O_B}) \tag{21}$$

Then, Eq. (19) can be made equivalent to Eq. (17) by setting

$$
\begin{aligned}
K_w &= K(\{(w + k)\Delta t\}_{k=\pm 0:\pm O_K}) \\
b &= \beta(\{k\Delta t\}_{k=\pm 1:\pm O_B})
\end{aligned}
\tag{22}
$$

which means $f \in \mathcal{G}$. Here concludes the proof that any CCNN layer can be matched by a CNN layer.

• $\mathcal{G} \subset \mathcal{F}$

Here we would only prove the case where both the weight network and bias network of CCNN has only one layer, which is the most difficult case. If either network has more than one layers, the proof is easily generalizable by setting the bottom layers to identity, which is feasible because of Cond. 4. Also, we only consider the case where the kernel network order and the bias network order are both ones, i.e.

$$
O_K = 1, O_B = 1 \tag{23}
$$

The proof can be generalized to larger orders by setting the additional weights to zero. In the special case defined above, the kernel network in Eq. (20) is further simplified to $K(w\Delta t)$. The bias network in Eq. (21) is further simplified to $\beta(0)$.

Further, notice that the $w\Delta t$, as the input to the kernel network, has to go through two-hot encoding. By Cond. 2 and 3, the two-hot encoding of $w\Delta t$ is a one-hot vector, where only the $w$-th dimension is activated and the rest is zero. Since the kernel network is a one-layer feedforward network, denote the weight connected to the $w$-th dimension of the two-hot vector as $P_w$, then we have

$$
K(w\Delta t) = P_w \tag{24}
$$

$\forall \{C_w\}$ and $b$ that define a CNN layer, and thereby $\forall g \in \mathcal{G}$, let

$$
P_w = \begin{cases} K_w & \text{if} - \frac{W-1}{2} \le w \le \frac{W-1}{2} \\ 0 & \text{otherwise} \end{cases} \tag{25}
$$

Cond. 3 ensures that there are enough number of nontrivial $P_w$s to cover all the $w$ in the case specified in line 1 of the equation above; and let

$$
\beta(0) = b \tag{26}
$$

which means the bias network learns a constant. Then the CCNN layer can be made equivalent to the CNN layer, i.e. $g \in \mathcal{F}$. Here concludes the proof that any CNN layer can be matched by a CCNN layer. □

Thm. 1 implies that in the uniform case where the increased model complexity of CCNN is not needed, it will not harm the performance either. Replacing CNN or CNNT with CCNN will not be worse off regardless of how the data are distributed in time.

As a final remark, readers may argue that the improved representation power of CCNN is merely a trivial result of increased model complexity, not because CCNN handles the time information more correctly. However, as will be shown in the next section, even with matching model complexity, CNN and CNNT are still unable to match the performance of CCNN. CCNN does not just increase the model complexity but increases the model complexity the right way.

## C  EXPERIMENT SETTINGS

### C.1  PREDICT FUTURE SIGNAL VALUE TASK: DATA GENERATION

Three standard time series functions are introduced as $x(t)$.

• *Sine:*

$$
x(t) = \sin\left(\frac{2\pi t}{T}\right) \tag{27}
$$

where $T = 5$. The sampling time intervals follow the Poisson distribution with mean parameter $\lambda = 1$.

Table 2: Mean squared error of prediction on realworld data.

| Alg. | DM1 | DM2 | DM3 | DM4 | DM5 | DM6 | DM7 | DM8 | DM9 | DM10 | DM11 | DM12 | DM13 |
|---|---|---|---|---|---|---|---|---|---|---|---|---|---|
| CNNT-th | 0.86 | 0.81 | 0.91 | 0.99 | 0.42 | 0.76 | 0.52 | 0.55 | 0.43 | 0.69 | 0.84 | 0.99 | 0.47 |
| ICNN-L | 0.52 | 0.27 | 0.70 | 1.03 | 0.06 | 0.30 | 0.02 | 0.24 | **0.26** | 0.44 | 0.51 | 0.84 | 0.13 |
| ICNN-Q | 0.61 | 0.28 | 0.69 | **0.98** | 0.06 | 0.30 | 0.06 | 0.23 | 0.29 | 0.45 | 0.57 | 0.84 | 0.13 |
| ICNN-C | 0.63 | 0.29 | 0.71 | 1.02 | 0.06 | 0.30 | 0.04 | 0.25 | 0.29 | 0.43 | 0.56 | 0.82 | 0.13 |
| ICNN-P | 0.71 | 0.40 | 0.74 | 0.99 | 0.09 | 0.37 | **0.02** | 0.28 | 0.33 | 0.45 | 0.60 | 0.94 | 0.17 |
| ICNN-S | 0.53 | 0.26 | 0.66 | 1.00 | 0.07 | 0.28 | 0.21 | 0.26 | 0.31 | 0.43 | **0.49** | 0.87 | 0.15 |
| RNNT-th | 0.79 | 0.27 | 0.70 | 1.06 | 0.05 | 0.38 | 0.42 | **0.20** | 0.33 | 0.48 | 0.57 | 1.01 | 0.11 |
| CCNN-th | **0.49** | **0.23** | **0.65** | 1.00 | **0.05** | **0.27** | 0.04 | 0.22 | 0.29 | **0.41** | 0.50 | **0.80** | **0.12** |

• *Mackey-Glass (MG):* a chaotic time series inspired by biological control systems (Glass & Mackey, 1979). MG is the solution to the following delay differential equation.

$$\dot{x}(t) = \beta \frac{x(t-\tau)}{1 + x(t-\tau)^n} - \gamma x(t) \tag{28}$$

where $\dot{x}$ denotes $dx/dt$. We choose $\beta = 0.2$, $\tau = 17$, $n = 10$, and $\gamma = 0.2$. There is no closed-form solution to Eq. (28), so the Runge-Kutta method Bogacki & Shampine (1989) is applied to obtain a discrete numerical solution, which is a uniform sequence with sampling interval $\Delta t$. A set of $N$-sample short uniform sequences are generated from the long sequence by a sliding window with window shift of one. The uniform sequences are subsampled into nonuniform sequences by choosing $M$ ($M < N$) sample points uniformly at random.[1] $\Delta t = 2$, $N = 42$, $M = 14$ on this dataset. The timestamps are normalized such that the expected interval of the nonuniform sequences is one.

• *Lorenz:* a simplified system describing the 2D flow of fluid (Lorenz, 1963). The Lorenz function is the solution to the following differential equation system.

$$\begin{aligned}
\dot{x}(t) &= \sigma(y(t) - x(t)) \\
\dot{y}(t) &= -x(t)z(t) + rx(t) - y(t) \\
\dot{z}(t) &= x(t)y(t) - bz(t)
\end{aligned} \tag{29}$$

We choose $\sigma = 10$, $r = 28$, $b = 8/3$. The nonuniform subsequences are generated the same way as in MG with $\Delta t = 0.05$. Only $x(t)$ is observed and predicted.

## C.2   PREDICT FUTURE SIGNAL VALUE TASK HYPERPARAMETERS

As mentioned, the hyperparameters are set such that all the architectures share the same number of layers, receptive field size, and number of parameters. All the hidden activation functions are ReLU, and all the output activation functions are linear.

**CNN:** CNN has two $1 \times 7$ layers, and the number of hidden channels is 84. The total number of parameters is 1,261.

**CNNT:** CNNT has two $1 \times 7$ layers, and the number of hidden channels is 60. The total number of parameters is 1,261. The sampling time intervals are appended as input features. The time interval appended to the last sample of the sequence is the difference between the prediction time and the time of the last sample.

**CNNT-th:** The two-hot encoding interval width $\delta = 0.5$, and the two-hot vector dimension is 14. The network has two $1 \times 7$ layers, and the number of hidden channels is 10. The total number of parameters is 1,261.

**ICNN:** ICNN takes the interpolated signal at $t_{n+1} - k$, $k = 1, \cdots, 13$ as input. The network has two $1 \times 7$ layers, and the number of hidden channels is 84. The total number of parameters is 1,261.

**RNNT:** RNNT and RNNT-th have two layers and the number of hidden channels is 32. The total number of parameters is 1153 for RNNT and 1633 for RNNT-th.

**CCNN:** CCNN has two layers. The first layer is a CCNN layer with output timestamps at $t_{n+1} - k$, $k = 1, \cdots, 13$. The bias network has two layers and 72 hidden channels. Its order $O_B$ is 7. The

---

[1]Equivalent to a Poisson process conditional on the event that the number of occurrences by the time $N$ is $M$.

kernel network has two layers and 4 hidden nodes. Its order $O_K$ is 3. The kernel length $L_k = 3$. The second layer is a regular $1 \times 7$ convolutional layer. The total number of parameters is 1,273.

**CCNN-th:** CCNN-th has almost the same structure as CCNN, except that the number of hidden channels is 36. The total number of parameters is 1,261.

### C.3 DESCRIPTION OF REAL-WORLD DATASET FOR PREDICTING INTERVALS TO NEXT EVENT

**Stackoverflow:** The dataset contains 26535 training samples, 3315 test samples, and 3315 validation samples. Each sequence represents the history awarding *badges* to a single user. The sampling timestamps are the time of awarding each badge, and the signal value is the type of the badge (e.g., Guru, Nice Question, Great Answer, etc.). There are 22 types of badges in total.

**Retweet:** The dataset contains 20000 training samples, 1000 test samples, and 1000 validation samples. The sequence is the history of a tweet being re-posted by other users. According to the number of followers of the users retweeted the post, the label of each retweet, in the whole retweet history, is one of the 3 the user groups.

**MIMIC:** The dataset is the collection of clinical visit history. The training set contains 2925 sequences; the test and validation set contains 312 each. The diagnosis are filtered to preserve only top-10 common diseases as the label.

**NYSE** The dataset is book order data collected from NYSE of high-frequency transaction in one day. The dataset contains 2 types of events (sell and buy) with 298710 training sequences, 33190 testing sequences, and 82900 validation sequences.

### C.4 CONFIGURATION OF PREDICTING TIME INTERVAL TO NEXT EVENT

The input sequence length to NYSE, Stackoverflow, and ReTweet dataset is 13, and the CCNN uses two $1 \times 7$ kernels with 16 filters for each. MIMIC contains only very short sequences, so CCNN uses two $1 \times 2$ kernels and predicts only based on past 3 events.

## D ADDITIONAL EXPERIMENTS

### D.1 PREDICT FUTURE SIGNAL VALUE ON REAL-WORLD DATASET WITH MISSING DATA

In order to test the advantage of CCNN in real-world scenarios with missing observations, 13 time-series datasets are randomly chosen from the Data Market[2], named DM1 through DM13. A brief description of these datasets is given below. Each dataset consists of a univariate uniform time series, which is split into training, test and validation sequences by a ratio of 6:2:2. Nonuniform subsequences are generated the same way as in MG with $N = 28$ and $M = 14$. All of the data are monthly data. The network configurations are the same as those in the simulated experiment. In particular, the receptive field size is set to seven, which means the prediction is based on an average of 14 months of historic data. This should lend adequate information for prediction.

**DM1:** Australia monthly production of cars and station wagons from Jul 1961 to Aug 1995.[3] The total length of the sequence is 414.

**DM2:** Monthly data on Clearwater River at Kamiah, Idaho from 1911 to 1965.[4] The total length of the sequence is 604.

**DM3:** Monthly data on James River at Buchanan, VA from 1911 to 1960.[5] The total length of the sequence is 604.

---

[2]https://datamarket.com

[3]https://datamarket.com/data/set/22lf/australia-monthly-production-of-cars-and-station-wagons-jul-1961-aug-1995#!ds=22lf&display=line

[4]https://datamarket.com/data/set/22zg/clearwater-river-at-kamiah-idaho-1911-1965#!ds=22zg&display=line

[5]https://datamarket.com/data/set/22y3/james-river-at-buchanan-va-1911-1960

**DM4:** Mean monthly precipitation from 1907 to 1972.[6] The total length of the sequence is 796.

**DM5:** Mean monthly temperature from 1907 to 1972.[7] The total length of the sequence is 796.

**DM6:** Monthly data on Middle Boulder Creek at Nederland, CO from 1912 to 1960.[8] The total length of the sequence is 592.

**DM7:** Monthly electricity production in Australia (million kilowatt hours) from Jan 1956 to Aug 1995.[9] The total length of the sequence is 480.

**DM8:** Monthly flows of Chang Jiang (Yangtze River) at Han Kou, China from 1865 to 1979.[10] The total length of the sequence is 1372.

**DM9:** Monthly production of clay bricks (million units) from Jan 1956 to Aug 1995.[11] The total length of the sequence is 480.

**DM10:** Monthly rain in Coppermine (mm) from 1933 to 1976.[12] The total length of the sequence is 532.

**DM11:** Monthly riverflow (cms) in Pigeon River near Middle Falls, Ontario from 1924 to 1977.[13] The total length of the sequence is 640.

**DM12:** Monthly riverflow (cms) in Turtle River near Mine Centre, Ontario from 1921 to 1977.[14] The total length of the sequence is 676.

**DM13:** Monthly temperature in England (F) from 1723 to 1970.[15] The total length of the sequence is 2980.

Table 2 shows the mean squared prediction errors. CCNN-th maintains its lead on most datasets. Where it does not, different ICNNs alternately take the lead by small margins. Here are two comments. First, note that the kernel length in ICNN-C and ICNN-Q is much larger than $L_K$, so it falls beyond the representation power of CCNN. Nevertheless, this result shows that an interpolation kernel within the scope of Eq. (1) usually suffices to outperform the standard interpolation methods that fall beyond. Second, unlike in the simulated test, ICNN-L generally performs better than ICNN-C, which emphasizes the importance of choosing a suitable interpolation scheme for each task. CCNN, with its ability to choose its own kernels, avoids such trouble.

### D.2 Speech Interpolation

Since CCNN is motivated by interpolation, it is insightful to see CCNN's performance in interpolation tasks. The speech interpolation task involves restoring the high-resolution speech signal from the downsampled signal.

### The Dataset

To mitigate the complexity in directly working on speech waveforms, the sampling rate of the high-resolution speech is set to 4 kHz, and that of the downsampled signals is 2 kHz. Three different downsampling schemes are tested. The first scheme, called *uniform filtered*, uniformly downsamples the speech to 2 kHz after passing it to an anti-aliasing filter. The second scheme, called *uniform unfiltered*, uniformly downsamples the signal without the anti-aliasing filter. The third scheme, called *nonuniform*, randomly preserves half of the speech samples, and thus the resulting signal has an average sampling rate of 2 kHz.

---

[6]https://datamarket.com/data/set/22w1/mean-monthly-precipitation-1907-1972

[7]https://datamarket.com/data/set/22o4/mean-monthly-temperature-1907-1972

[8]https://datamarket.com/data/set/22vt/middle-boulder-creek-at-nederland-co-1912-1960

[9]https://datamarket.com/data/set/22l0/monthly-electricity-production-in-australia-million-kilowatt-hours-jan-1956-aug-1995

[10]https://datamarket.com/data/set/22r8/monthly-flows-chang-jiang-at-han-kou-1865-1979

[11]https://datamarket.com/data/set/22lv/monthly-production-of-clay-bricks-million-units-jan-1956-aug-1995

[12]https://datamarket.com/data/set/22n8/monthly-rain-coppermine-mm-1933-1976

[13]https://datamarket.com/data/set/22mi/monthly-riverflow-in-cms-pigeon-river-near-middle-falls-ont-1924-1977

[14]https://datamarket.com/data/set/22mf/monthly-riverflow-in-cms-turtle-river-near-mine-centre-ont-1921-1977

[15]https://datamarket.com/data/set/22vp/monthly-temperature-in-england-f-1723-1970

Table 3: Signal-to-Noise Ratio (dB) in Speech Interpolation.

| Alg. | Uniform | | Non-uniform |
|---|---|---|---|
| | filtered | non-filtered | |
| Speech DNN | 9.13 | -* | -* |
| Speech DNN (CP) | **13.64** | -* | -* |
| ICNN-C | 9.74 | 7.49 | 2.33 |
| ICNN-Q | 9.66 | 5.67 | 2.77 |
| ICNN-L | 9.72 | 5.81 | 3.16 |
| ICNN-P | 3.17 | 2.63 | 1.82 |
| ICNN-S | 9.62 | 5.90 | 2.83 |
| CCNN-th | 9.61 | **7.80** | **6.58** |

* Speech DNN does not work with non-filtered down-sampled signals and nonuniformly down-sampled signals.

Our dataset consists of one hour of lab-recorded speech of one speaker reading structured composite sentences. We use 80% of the dataset as training, 10% as validation, and the rest of 10% as test. The high-resolution speech is chunked into 40-point sequences without overlap, and the corresponding downsampled speech into 20-point sequences.

CONFIGURATIONS

Similar to the prediction experiment, the ICNN approaches interpolates the low-resolution speech into high-resolution speech (4 kHz) before it is fed to a two-layer regular CNN. A similar practice has also been adopted in Kim et al. (2016). CCNN also has two layers. The first layer outputs at the uniform timestamps at the rate of 4 kHz, and the second layer is a regular convolutional layer. Detailed configurations are shown below. Again, the hyperparameters are set such that the two architectures have the same number of layers, receptive field size and model complexity. The activation functions for CCNN and ICNN are both sigmoid. The hyperparameters are detailed below.

**CCNN-th:** The timestamps are normalized such that the sampling interval of the original 4 kHz speech is 0.5. CCNN-th has two layers. The first layer is a CCNN layer that outputs the timestamps uniformly distributed at the rate of 4 kHz. The number of hidden channels is 16. The bias network has one layer, and the order $O_B = 28$. The bias kernel network has one layer, and the order $O_K = 7$. The kernel length $L_K = 10$. The two-hot encoding time interval $\delta = 0.5$. The second layer is a $1 \times 28$ convolutional layer.

**ICNN-C:** The CNN contains two $1 \times 28$ convolutional layers with 80 number of hidden filters.

**Speech DNN & Speech DNN CP:** Speech DNN converts both the high resolution and downsampled speech into amplitude spectra using FFT with 64ms window length and 48ms window shift. Speech DNN has 3 hidden layers, each of which has 1024 hidden nodes. Because our temporal resolution is half of that in Li & Lee (2015), our hidden node number is a half of that in Li & Lee (2015) too. The number of parameters in Speech DNN is around $2.3 \times 10^6$, which is much larger than that in CCNN and the ICNN baseline models (both have around $5 \times 10^3$ parameters).

Since sample-based speech interpolation methods have yet to achieve the state-of-the-art, we also include a spectrum-based benchmark from Li & Lee (2015) called Speech DNN. Speech DNN only works on the uniform filtered case. It predicts the higher half of the amplitude spectrum of the high-resolution speech of that of the low-resolution speech. The flipped phase spectrum of the down-sampled speech is used as an estimate of the higher half of the phase spectrum of the high-resolution speech.

Since the phase spectrum estimate of Speech DNN can be inaccurate, we introduced another version, called Speech DNN with a cheated phase (CP), where the ground truth phase spectrum is applied. Note that this version is given too much oracle information to be fairly compared with. Nevertheless, we will include its results for reference.

RESULTS AND ANALYSIS.

Tab. 3 shows the Signal-to-Noise Ratio (SNR) of the signals recovered from different input signals by different models. The Speech DNN with cheated phase yields the best SNR, because it uses the phase information from ground truth. However, the Speech DNN without cheated phase has similar

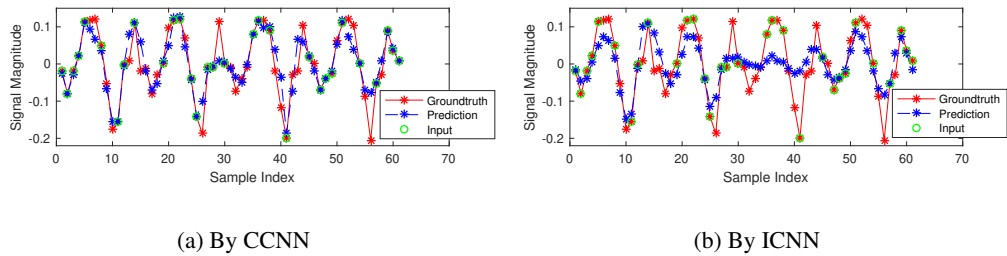

(a) By CCNN                                    (b) By ICNN

Figure 7: Examples for CCNN and ICNN in restoring nonuniformly down-sampled speech. CCNN generates better approximation at especially at crests and troughs.

performance to CNN and CCNN, even though it has much more weights, largely because of the inaccurate phase estimates.

As for the comparison between CCNN and ICNN, they have similar SNR under uniform sampling cases. This verifies that both architectures have similar representation power given uniform data. However, CCNN has much higher SNR than ICNN in nonuniform case. One important reason is that CCNN, by construction, is aware of whether neighboring samples are dense or sparse, and outputs robust interpolation kernels accordingly, despite the variation in sampling patterns; whereas CNN is unable to deal with various random sampling instances.

Fig. 7 shows an example of the restored signal from the nonuniform samples by CCNN and CNN respectively. CCNN can restore some spikes (*e.g.* ones at 25 and 40) even without an input sample point in the spike, because CCNN can learn the continuous kernels and restore the original spikes. CNN model does poorly in restoring spikes even when there are input sample points in the spikes.

