# OpenReview forum: "Continuous Convolutional Neural Network forNonuniform Time Series"
_ICLR.cc/2020/Conference — Reject_

### Official Review · AnonReviewer2 · 2019-10-23
**Official Blind Review #2**

**Rating:** 3

**Review:**

A method that was proposed by authors deals with a problem of non-uniform data in time series. One of ways to deal with this problem is interpolate input signal between data points. In signal processing a standard way to interpolate is to apply a convolution with a kernel. This operation, by itself, is a non-trivial, since we lack any information about signal's spectrum (and do not know optimal kernel). Thus, the authors propose to search for a kernel in a form neural network. To this term they also add bias term which is also a neural network.

Basic idea of the paper seems promising, but reported results are only partial. Since a paper is experimental, i.e. no theory at all, then the main judgement should be based on experimental results. They are not convincing, as CCNN is compared with methods that can not be called state-of-the-art.

**Experience Assessment:**

I have read many papers in this area.

**Review Assessment: Checking Correctness Of Derivations And Theory:**

I assessed the sensibility of the derivations and theory.

**Review Assessment: Checking Correctness Of Experiments:**

I assessed the sensibility of the experiments.

**Review Assessment: Thoroughness In Paper Reading:**

I made a quick assessment of this paper.

---

> ### Author Response · Authors · 2019-11-15
> **Response to Reviewer 2**
>
> Thank you for your feedback! Below is our response to your concern.
>
> 1. Experimental v.s. theoretical
> It’s hard to agree with the claim of the reviewer that a paper is experimental. If the reviewer means particularly this paper is purely theoretical, I would apologize for not emphasizing enough to let the reviewer see the continuous convolution theory in section 3 and the representation power proof in appendix B.
>
> 2. Experimental result and state-of-the-art
> On the one hand, we showed the performance of CCNN together with two other baselines in figure 5. Comparing to the marginal improvement of RMTPP from its baseline N-SM-MPP, the performance gain of CCNN is actually significant. On the other hand, we honestly believe that RMTPP and N-SM-MPP are current state-of-the-art of the task.

---

### Official Review · AnonReviewer3 · 2019-10-23
**Official Blind Review #3**

**Rating:** 3

**Review:**

The paper proposes a continuous CNN model to accommodate the nonuniform time series data. The model learns the interpolation and the convolution kernel functions in an end-to-end manner, so that it can capture the signal patterns and be flexible. A layer has three networks, which learn a kernel function to represent the combination of interpolation and convolution, a bias function to represent the error correction with convolution, and then produce the output based on them. The authors introduce two assumptions and a two-hot encoding scheme for the input to control the model complexity. The paper also introduces an application of the proposed CCNN by combing with temporal point process. Experiments on simulated data compares the proposed method with some degenerative baselines show the advantage of learning the interpolation and the two-hot encoding configuration. The authors compare the performance on time interval prediction task based on real world dataset to show the model produces a better history embedding for the task.
Overall, the paper has some incremental improvements on the existing methods that dealing with the nonuniform time series data. Instead of using preset interpolation kernels, the proposed model can learn it with the convolution in a data-driven manner.
The paper includes clear explanation on module structure and detailed experiment settings.
The experiments of signal value prediction support the claims of the advantages of the proposed model.
The notation in the caption of figure 1 is a little confusing: is x(t_i’) the same as hat x(t) in the algorithm?
It is good that the related works section mentioned the adapted RNNs that are used as baselines in the real-world dataset experiment, and the differences between the proposed model and the related SNNs are introduced.
However, this section and the introduction can be better organized to distinguish the novelty and the contribution of the work.
In page 7, the purpose of the reference in the sentence “The time information is either two-hot encoded (Adams et al., 2010) (CCNN-th), or not encoded (CCNN).” is not very clear.
It will be better if there are a little more analysis of the experiment on predicting time intervals to next event.
The upper plots in the figure may be not convincing enough to support the claims.
The advantage of two-hot encoding seems subtle in the figure 5. Is there any reason for the significantly higher deviation of CCNN-th in StackOverflow?
Sometimes the usage of “CCNN” is not clear, for example, the experiment on speech interpolation compares the “CCNN-th” method with baselines, but uses “CCNN” in the analysis. Also, it could be better to show the “CCNN-th” result in the upper plots of figure 5 instead of “CCNN”.
Minor comment:
There are some typos in the paper, for example, missing the right parenthesis in page3 “(refer to Appendix A.1”, in page4 section 4.1 “According to Eq.(4), the input is …”.
“The left plot shows” in the last line of the caption of figure 3 should be “right”.

**Experience Assessment:**

I have read many papers in this area.

**Review Assessment: Checking Correctness Of Derivations And Theory:**

I assessed the sensibility of the derivations and theory.

**Review Assessment: Checking Correctness Of Experiments:**

I assessed the sensibility of the experiments.

**Review Assessment: Thoroughness In Paper Reading:**

I read the paper at least twice and used my best judgement in assessing the paper.

---

> ### Author Response · Authors · 2019-11-15
> **Response to Reviewer 3**
>
> We appreciate the reviewer’s thorough and careful reading of the manuscript and the feedback. We have fixed typos in the revised submission, and below is our response to other concerns:
>
> 1. The notation in the caption of figure 1 is a little confusing: is x(t_i’) the same as hat x(t) in the algorithm?
> Thanks for pointing this out. Yes, they are the same. We’ve changed the notation in the caption to make it consistent with the content.
>
> 2. The upper plots in the figure may be not convincing enough to support the claims.
> We admit the prediction on the upper plots of figure 5, especially the NYSE, shows the gap between prediction and ground truth data. However, we should agree that the prediction on stock transaction in real-world is fairly challenging and there’s no evidence showing how predictable is the interval to the next transaction. We could have shown only pretty result on the remaining dataset, but we decided to show the result of NYSE, as it could be a good demonstration of improvement of CCNN on very challenging task.
>
> 3. The advantage of two-hot encoding seems subtle in figure 5. Is there any reason for the significantly higher deviation of CCNN-th in StackOverflow?
> Our experiment controls that all architectures have the same number of hyperparameters. Since the two hot encoding requires more hyperparameters at the input layers, we have reduced the filter size to match the number of hyperparameters to the other architectures. That is why the performance is sometimes compromised.

---

### Official Review · AnonReviewer1 · 2019-10-25
**Official Blind Review #1**

**Rating:** 3

**Review:**

1.	The motivation of continuous convolution is not very clear, can the authors please motivate? To my understanding this is just to handle inputs with unequal time steps, but that can be handled multiple ways, why not just naively resample?
2.	The proposed network was defined as continuous convolution followed by the standard convolution. Why not just stack multiple continuous convolutions?
3.	Continuous convolution should be a general case for standard convolution, can authors explicitly show it?
4.	Another way to handle unequal timesteps is by using dilated convolution, can authors please comment how they differ, pros and cons etc.?
5.	Two hot encoding seems another way to discretize, no?
6.	The experiments section is rather weak, CCNN seems to have a lot of spikes in prediction, e.g., in Fig. 5.
7.	It’s very strange why two hot encoding does not perform that well, while reading the method section, it seems very obvious to take two ends of an interval, in that way two hot encoding seems logical.

Overall it seems like an easy extension with a lot of parts not well-justified. Also I don't clearly have a well-grounded motivation for a continuous convolution.


**Experience Assessment:**

I have published one or two papers in this area.

**Review Assessment: Checking Correctness Of Derivations And Theory:**

I did not assess the derivations or theory.

**Review Assessment: Checking Correctness Of Experiments:**

I assessed the sensibility of the experiments.

**Review Assessment: Thoroughness In Paper Reading:**

I made a quick assessment of this paper.

---

> ### Author Response · Authors · 2019-11-15
> **Response to Reviewer 1**
>
> Thank you for your feedback. We have noticed that there might be some misunderstandings of the concept of a non-uniform time series. We hope our response below will help clarify these misunderstandings.
>
> 1. Motivation of CCNN
> Our target applications are non-uniform time series, where the time intervals between any adjacent timestamps can be different. For example, a possible set of time stamps could be {0, 0.01, 5, 7.23, 11.1...}. For such non-uniformly-sampled signals, resampling requires first interpolating the signal. As we have shown in the experiment in section 5.1 and appendix D, the selection of interpolation kernels has a direct influence on the performance, and applying any existing preset interpolation kernels can lead to a compromise in the performance. Motivated by this observation, we proposed the end-to-end method to learn the interpolation kernel in a data-driven way, to eliminate the need of trying through preset interpolation kernels and to overcome the challenge when the underlying interpolation for some time-series is unknown.
>
> Please note that we are not directly applying continuous convolution, but rather use continuous convolution, together with the sampling theory, to derive the basic CCNN operation, as shown in section 3. The reason why continuous convolution is indispensable in this derivation is that its convolution kernel is well-defined on any possible time stamps, and so it can align with the non-uniform timestamps in the input signal.
>
> 2. Stacked CCNN v.s. CCNN + standard CNN
> Continuous convolution layers can be stacked, but it is necessary only when the output timestamps in the previous layers are non-uniform. If the output time intervals of the intermediate CCNN layer are uniform, then applying standard CNN and CCNN in the subsequent layers are equivalent.
>
> 3. Strict generalization to standard CNN
> Appendix B provides a theoretical analysis of the relation between standard CNN and CCNN. In short, when the input timestamps and output timestamps are both uniformly sampled, CCNN would be equivalent to standard CNN. Please refer to Appendix B for the formal statement of the theorem and the proof.
>
> 4. CCNN v.s. Dilated CNN
> These two methods have fundamental differences on assumption. Dilated CNN deals with multi-resolution situation, in which for each scale the signal should still be uniformly sample but of difference resolution. CCNN releases the uniform sampling assumption and requires neither the input signal being uniformly sampled nor the intervals sharing some common factor (which could be regarded as finest resolution).
>
> 5. Two hot encoding seems another way to discretize, no?
> Yes
>
> 6. Experiment Result
> CCNN did have spikes in prediction in figure 5, which indeed confirmed its good performance on that task. The y axis of the line chart in figure 5 represents the time interval to the next event, and a spike indicates some events happen after a long time interval from the previous one. We can see the spike of prediction aligns with the spikes of ground truth data, which is the intuitive evidence of CCNN learned the pattern of the temporal point process. Meanwhile, the quantitative performance evaluation also shows its advantage over existing methods on all of the four datasets.
>
> 7. Why two hot encoding does not perform that well
> Our experiment controls that all architectures have the same number of hyperparameters. Since the two hot encoding requires more hyperparameters at the input layers, we have reduced the filter size to match the number of hyperparameters to the other architectures. That is why the performance is sometimes compromised.

---

### Decision · Program_Chairs · 2019-12-19

**Decision:**

Reject

**Comment:**

This paper presents a continuous CNN model that can handle nonuniform time series data. It learns the interpolation kernel and convolutional architectures in an end-to-end manner, which is shown to achieve higher performance compared to naïve baselines.
All reviewers scored Weak Reject and there was no strong opinion to support the paper during discussion. Although I felt some of the reviewers’ comments are missing the points, I generally agree that the novelty of the method is rather straightforward and incremental, and that the experimental evaluation is not convincing enough. Particularly, comparison with more recent state-of-the-art point process methods should be included. For example, [1-3] claim better performance than RMTPP. Considering that the contribution of the paper is more on empirical side and CCNN is not only the solution for handing nonuniform time series data, I think this point should be properly addressed and discussed. Based on these reasons, I’d like to recommend rejection.

[1] Xiao et al., Modeling the Intensity Function of Point Process via Recurrent Neural Networkss, AAAI 2017.
[2] Li et al., Learning Temporal Point Processes via Reinforcement Learning, NIPS 2018.
[3] Turkmen et al, FastPoint: Scalable Deep Point Processes, ECML-PKDD 2019.